# Thinking Outside the Ball: Optimal Learning with Gradient Descent for Generalized Linear Stochastic Convex Optimization

**Idan Amir**
Tel Aviv University
idanamir@mail.tau.ac.il

**Roi Livni**
Tel Aviv University
rlivni@tauex.tau.ac.il

**Nathan Srebro**
Toyota Technological Institute at Chicago
nati@ttic.edu

## Abstract

We consider linear prediction with a convex Lipschitz loss, or more generally, stochastic convex optimization problems of generalized linear form, i.e. where each instantaneous loss is a scalar convex function of a linear function. We show that in this setting, early stopped Gradient Descent (GD), without any explicit regularization or projection, ensures excess error at most $\varepsilon$ (compared to the best possible with unit Euclidean norm) with an optimal, up to logarithmic factors, sample complexity of $\tilde{O}(1/\varepsilon^2)$ and only $\tilde{O}(1/\varepsilon^2)$ iterations. This contrasts with general stochastic convex optimization, where $\Omega(1/\varepsilon^4)$ iterations are needed Amir et al. [2]. The lower iteration complexity is ensured by leveraging uniform convergence rather than stability. But instead of uniform convergence in a norm ball, which we show can guarantee suboptimal learning using $\Theta(1/\varepsilon^4)$ samples, we rely on uniform convergence in a distribution-dependent ball.

## 1 Introduction

The recent success of learning using deep networks, with many more parameters than training points, and even without any explicit regularization, has brought back interest in how biases and "implicit regularization" due to the optimization method used, can ensure good generalization, even in situations where minimizing the optimization objective itself cannot [28]. Alongside interest in understanding the algorithmic biases of optimization in non-convex, deep models, and how they can yield good generalization [e.g. 3, 6, 8, 13–15, 21, 23–25, 29, 30, 39], there has also been renewed interest in understanding the fundamentals of algorithmic regularization in convex models [1, 2, 9, 10, 31, 33], both as an interesting and important setting in its own right, and even more so as a basis for understanding the situation in more complex models (how can we hope to understand phenomena in deep, non-convex models, if we can't even understand them in convex models?). In particular, these fundamental questions include the relationship between algorithmic regularization, explicit regularization, uniform convergence and stability; and the importance of stochasticity and early stopping.

In this paper we focus on the algorithmic bias of *deterministic* (full batch) optimization methods, and in particular of full-batch gradient descent (i.e. gradient descent on the empirical risk, or "training error"). Even when gradient descent (GD) is run to convergence, it affords some bias that can ensure generalization. E.g. in linear regression (and even slightly more general settings) where interpolation is possible, it can be shown to converge to the minimum norm interpolating solution. This can be

sufficient for generalization even in underdetermined situations where other interpolators (i.e. other minimizers of the optimization objective) would not generalize well. Indeed, the generalization ability of the minimum norm interpolating solution, and hence of GD, even in noisy situations, is the subject of much study. And in parallel, there has also been work going beyond GD on linear regression, characterizing the limit point of GD, or of other optimization methods such as Mirror Descent, steepest descent w.r.t. a norm, coordinate descent and AdaBoost, for different types of loss functions [12, 17–19, 22, 34, 35, 37, 38].

But what about situations where interpolation, or completely minimizing the empirical risk, is not desirable, and generalization requires compromising on the empirical risk? In this cases one can consider early stopping, i.e. running GD only for some specific number of iterations. Indeed, early stopped GD is a common regularization approach in practice, and other learning approaches, most prominently Boosting, can also be viewed as early stopping of an optimization procedure (coordinate descent in the case of Boosting). When and how can such early stopped GD allow for generalization? How does this compare to Stochastic Gradient Descent (SGD), or to using explicit regularization, both in terms of generalization ability, and the number of required optimization iterations? And what tools, such as uniform convergence, distribution-dependent uniform convergence, and stability, are appropriate for studying the generalization ability of GD?

**Our main result** We will show that when training a linear predictor with a convex Lipschitz loss (or more generally, for stochastic convex optimization with a generalized linear instantaneous objective), GD with early stopping *can* generalize optimally, up to logarithmic factors, with optimal sample complexity $\tilde{O}(1/\varepsilon^2)$, *and* with an optimal number $\tilde{O}(1/\varepsilon^2)$ of iterations (the same as stochastic gradient descent), even without any explicit regularization, and in particular *without* projections into a norm ball—just unconstrained GD on the training error. This contrasts to previous results regarding early stopped GD for arbitrary stochastic convex optimization (not necessarily generalized linear), for which Amir et al. [2] showed $\Theta(1/\varepsilon^4)$ GD iterations are needed (even if projections or explicit regularization is used).

**Stability vs Uniform Convergence for SGD and GD** The important difference here, and the only property of generalized linear models (GLMs) that we rely on, is that GLMs satisfy uniform convergence Bartlett and Mendelson [4]: empirical losses converge to their expectations uniformly over all predictors in Euclidean balls. This is in contrast to general Stochastic Convex Optimization (SCO), for which no such uniform convergence is possible [33]. Consequently, rather then uniform convergence, the analysis for SCO is based on stability arguments Bassily et al. [5]. For SGD, stability at each step can be used in an online analysis, combined with an online-to-batch conversion, which is sufficient for ensuring optimal generalization with an optimal $O(1/\varepsilon^2)$ number of iterations. But for GD, we must consider the stability of the method as a whole, which leads to optimal rates in the case of smooth losses Hardt et al. [16] but otherwise is much worse, necessitating using a smaller stepsize, and hence quadratically more iterations—[2] showed that this is not just an analysis artifact, but a real limitation of GD for general SCO. In this paper, we show that once generalization can be ensured via uniform convergence, e.g. for GLMs, then GD does not have to worry about stability, can take much larger step sizes, and generalize optimally after only $\tilde{O}(1/\varepsilon^2)$ iterations.

But we must be careful with how we ensure uniform convergence! To rely on uniform convergence, we need to ensure the output of GD lies in some ball, and the generalization error would then scale with the radius of the ball. A naïve approach would be to ensure that output of GD lies in a norm ball around the origin, and rely on uniform convergence in this ball. In Section 4 we show that this approach can ensure generalization, but only with suboptimal sample complexity of $O(1/\varepsilon^4)$—that is, worse than with the stability-based approach. Instead, we show that, with high probability, the output of GD lies in a small (constant radius) *distribution dependent* ball, centered not at the origin, but around the (distribution dependent) output of GD on the population objective. Even though this ball is unknown to the algorithm, this is still sufficient for generalization. The situation here is similar to the notion of algorithm dependent uniform convergence introduced by [26], though we should emphasize that here we show that algorithm-dependent uniform convergence *is* sufficient for optimal generalization.

**Context and Insights** Our results complement recent results exposing gaps between generalization using stochastic optimization versus explicit regularization or deterministic optimization in stochastic convex optimization. [31] showed that for a setting that is slightly weaker (where only the population

loss is convex, but instantaneous losses can be non-convex), there can be large gaps between SGD versus GD or explicit regularization: even though SGD can learn with the optimal sample complexity of $O(1/\varepsilon^2)$, explicit regularization, in the form of regularized empirical risk minimization cannot learn at all, even with arbitrarily many samples, and GD requires at least a suboptimal $\Omega(1/\varepsilon^{2.4})$ number of samples (it is not clear if this sample complexity is sufficient for GD, or even if it can at all ensure learning). This highlights that the generalization ability of SGD *cannot* be understood in terms of mimicking some regularizer, as well as gaps between stochastic and deterministic optimization.

Returning to the strict SCO setting, where the instantaneous losses are convex, still, uniform convergence does not hold, and generalization can only be ensured via algorithmic-dependent bounds. Nevertheless, optimal generalization can be ensured either thorough explicit regularization, SGD or GD—all three approaches can ensure learning with $\Theta(1/\varepsilon^2)$ samples [5, 27, 33]. Even so, differences between deterministic and stochastic methods still exist. [9] shows that in SCO, the output of SGD cannot even be guaranteed to lie in some "small" *distribution-dependent* set of (approximate) empirical risk minimizers[1], and so the generalization ability of SGD in this settings *cannot* be attributed to any "regularizer" being small. Dauber et al. also shows that GD does not follow some *distribution-independent* regularization path. In contrast, here we show that if we do take the distribution into account, then GD is constrained to follow (at least approximately) a predetermined path (i.e. deterministic path, that is independent of the sample, but does depend only on the distribution). Finally, there is also a gap between SGD and GD, in this case, in the required number of iterations [2]. Surprisingly, this gap cannot be fixed by adding regularization to the objective [1].

Importantly, for either weak or strict SCO, regularization in the form of constraining the norm of the predictor (i.e. empirical risk minimization inside the hypothesis class we are competing with), is *not* sufficient for learning [10, 33]. The failure of constrained ERM is critical here for the constructions of Amir et al., Dauber et al. and Sekhari et al. establishing the gaps above: since projected gradient descent would quickly converge to the constrained ERM, it would also generalize just as well, and the gaps and constructions are valid also for projected gradient descent.

In this paper we turn to the GLM setting, which is perhaps one of the most well studied frameworks in the learning theory literature, as it captures fundamental problems such as logistic regression, SVM and many more. Uniform convergence for GLMs [4, 20, 32] ensures that constrained ERM learns with optimal sample complexity, and hence so would *projected* GD with $O(1/\varepsilon^2)$ iterations. Interestingly, GD on the Tikhonov-type regularized objective similarly yields optimal learning with only $O(1/\varepsilon^2)$ iterations [36], indicating the lower bounds of Amir et al. on the iteration complexity do not hold in this setting. But both projected GD and GD on the Tikhonov regularized objective are forms of explicit regularization, and so we study unregularized, *unprojected*, GD. Our results show that (a) when uniform convergence holds, both sample complexity and iteration complexity gaps between stochastic and deterministic optimization disappear (though stochastic optimization is of course still much more computationally efficient), and (b) perhaps surprisingly, *distribution-dependent* uniform convergence is not only able to explain generalization of the output of GD, but it can do so better than stability; (c) though it appears to be suboptimal, distribution independent uniform convergence can explain generalization even though the norm could become very large (by making explanations based on uniform convergence inside a fixed distribution independent class).

## 2 Problem Setup and Background

We study the problem of stochastic optimization from i.i.d. data samples. For that purpose, we consider the standard setting of stochastic convex optimization. A learning problem consists of a family of loss functions $f : \mathcal{W} \times \mathcal{Z} \to \mathbb{R}$ defined over a fixed domain $\mathcal{W} \subseteq \mathbb{R}^d$, parameterized by the parameter space $\mathcal{Z}$. For each $B$ we denote the domain

$$\mathcal{W}^B = \{w \in \mathbb{R}^d : \|w\| \le B\}.$$

Our underlying assumption is that for each $z \in \mathcal{Z}$ the function $f(w; z)$ is convex and $L$-Lipschitz with respect to its first argument $w$. In this setting, a learner is provided with a sample $S = \{z_1, \ldots, z_n\}$ of i.i.d. examples drawn from an unknown distribution $D$. The goal of the learner is to optimize the

---

[1]"smallness" here, refers to a size notion that measures how well an empirical risk minimizer over the set generalizes, see [9] for the definition of a *statistically complex set*

*population risk* defined as follows:

$$F_D(w) := \mathop{\mathbb{E}}_{z \sim D} [f(w; z)].$$

More formally, an algorithm $\mathcal{A}$ is said to learn the class $\mathcal{W}$ (in expectation), with sample complexity $m(\varepsilon)$ if given i.i.d. sample $S = \{z_1, \ldots, z_n\}$ such that $n \geq m(\varepsilon)$, the learner returns a solution $\mathcal{A}(S)$ that holds

$$\mathop{\mathbb{E}}_{S \sim D^n} [F_D(\mathcal{A}(S))] \leq \min_{w \in \mathcal{W}} F_D(w) + \varepsilon.$$

A prevalent approach in stochastic optimization, is to consider, and optimize, the *empirical risk* over a sample $S$:

$$F_S(w) = \frac{1}{n} \sum_{i=1}^{n} f(w; z_i). \tag{1}$$

## 2.1 Gradient Descent (without projections).

A concrete and prominent way in minimizing the empirical risk in Eq. (1) is with *Gradient Descent* (GD). GD is an iterative algorithm that runs for $T$ iterations, and has the following update rule that depends on a *learning-rate* $\eta$:

$$w_{t+1}^S = w_t^S - \eta \nabla F_S(w_t^S), \tag{2}$$

where $w_0^S = 0$, and $\nabla F_S(w_t^S)$ is the (sub)gradient of $F_S$ at point $w_t^S$. The output of GD is normally taken to be the average iterate,

$$\bar{w}^S = \frac{1}{T} \sum_{t=1}^{T} w_t^S. \tag{3}$$

Throughout the paper we consider the aforementioned output in Eq. (3), though we remark that our results also extend to any reasonable averaging scheme (including prevalent schemes such as tail-averaging or random choice of $w_t$).

It is well known (e.g. [32]) that the output of GD with standard initialization at the origin, enjoys the following guarantee over the *empirical risk*, for every $B$:

$$F_S(\bar{w}^S) \leq \min_{w \in \mathcal{W}^B} F_S(w) + \eta L^2 + \frac{B^2}{\eta T}. \tag{4}$$

In contrast, as far as the population risk, it was recently shown in [2] that, for sufficiently large $d$, GD suffers from the following sub optimal rate[2]

$$\mathop{\mathbb{E}}_{S \sim D^n} [F_D(\bar{w}^S)] \geq \min_{w \in \mathcal{W}^1} F_D(w) + \Omega\left(\min\left\{\eta \sqrt{T} + \frac{1}{\eta T}, 1\right\}\right). \tag{5}$$

In particular, setting for concreteness $B = 1$, a choice of $\eta = 1/\sqrt{T}$ and $T = O(1/\varepsilon^2)$ may lead to $\varepsilon$-error over the empirical risk, but is susceptible to overfitting. In fact, it turns out that $T = O(1/\varepsilon^4)$ iterations are necessary and sufficient [5] for GD (with or without projections) to obtain $O(\varepsilon)$ population error.

## 2.2 Generalized Linear Models.

One, important, class of convex learning problems is the class of *generalized linear models* (GLMs). In a GLM problem, $f(w; z)$ takes the following generalized linear form:

$$f(w; z) = \ell(w \cdot \phi(x), y), \tag{6}$$

where $\mathcal{Z} = \mathcal{X} \times \mathcal{Y}$, $\ell(a, y)$ is convex and $L$-Lipschitz w.r.t. $a$, and, $\phi : \mathcal{X} \to H$ is an embedding of the domain $\mathcal{X}$ into some norm bounded set in a linear space (potentially infinite).

As we mostly care about norm bounded solutions, we will assume for concreteness that $\|\phi(x)\| \leq 1$. We also treat the value of the function at zero as constant, namely $|\ell(0, y)| \leq O(1)$. In turn, the function $f$ is $L$-Lipschitz as well, and we obtain by Lipschitzness that:

$$|f(w, z)| = |\ell(w \cdot \phi(x), y)| \leq O(\|w\| L). \tag{7}$$

---

[2]the result in [2] is formulated for projected-GD, however as the authors note the proof holds verbatim to the unprojected version.

**Uniform Convergence of GLMs.** One desirable property of GLMs is that, in contrast with general convex functions, they enjoy *dimension-independent uniform convergence bounds*. This, potentially, allow us to circumvent the bound in Eq. (5). In more detail, a seminal result due to Bartlett and Mendelson [4] shows that, under our restrictions on $\ell$ and $\phi$, the empirical risk $F_S(w)$ converges uniformly to the population loss $F_D(w)$ as follows for any $B$:

$$\mathop{\mathbb{E}}_{S \sim D^n} \Big[ \sup_{w \in \mathcal{W}^B} \big\{ F_D(w) - F_S(w) \big\} \Big] \leq \frac{2LB}{\sqrt{n}}. \tag{8}$$

By a similar reasoning, one can show a stronger property for GLMs, which we will need: uniform convergence on *any* ball, not necessarily centered around zero. More formally, for every $u$ and $K$ let us notate

$$\mathcal{W}_u^K := \{ w : \|w - u\| \leq K \}.$$

We can then bound the expected generalization error of $w \in \mathcal{W}_u^K$ as follows:

**Lemma 2.1.** *Suppose $\ell(a, y)$ is $L$-Lipschitz w.r.t. $a$ and $\phi : \mathcal{X} \to H$ is an embedding in a linear space $H$ such that $\|\phi(x)\| \leq 1$. Then, for $f$ as in Eq. (6). We have for any $u$ and $K$*

$$\mathop{\mathbb{E}}_{S \sim D^n} \Big[ \sup_{w \in \mathcal{W}_u^K} \big\{ F_D(w) - F_S(w) \big\} \Big] \leq \frac{2LK}{\sqrt{n}}$$

The proof is essentially the same as the proof of Eq. (8), and exploit the Rademacher complexity of the class. For the sake of completeness we repeat it in the supplementary material.

Eq. (8), and more generally Lemma 2.1, imply that *any* algorithm, $\mathcal{A}$ whose range is restricted to a $B$-bounded ball, that has an optimization guarantee of,

$$F_S(\mathcal{A}(S)) - \min_{w \in \mathcal{W}^B} F_S(w) \leq O\Big( \frac{LB}{\sqrt{n}} \Big),$$

will also obtain an $O(LB/\sqrt{n})$ convergence rate of the excess population risk. For example, projected-GD is an algorithm that enjoys the aforementioned generalization bound.

## 3 Main Results

**Theorem 3.1.** *Suppose $D$ is a distribution over $\mathcal{Z} = \mathcal{X} \times \mathcal{Y}$. Let $f : \mathbb{R}^d \times \mathcal{Z} \to \mathbb{R}$ be a loss function of the form given in Eq. (6) such that for all $y \in \mathcal{Y}$, $\ell(\cdot, y)$ is convex and $L$-Lipschitz with respect to its first argument and such that for all $x \in \mathcal{X} : \|\phi(x)\| \leq 1$. Then, for the gradient descent solution in Eq. (3) and for any $B \in \mathbb{R}$ we have:*

$$\mathop{\mathbb{E}}_{S \sim D^n} \big[ F_D(\bar{w}^S) \big] - \min_{w \in \mathcal{W}^B} F_D(w) \leq \tilde{O}\Big( \frac{\eta L^2 T}{n} + \frac{\eta L^2 \sqrt{T}}{\sqrt{n}} + \eta L^2 + \frac{B^2}{\eta T} \Big).$$

*In particular, setting $\eta = O(1/(L\sqrt{T}))$ and $T = n$ we get,*

$$\mathop{\mathbb{E}}_{S \sim D^n} \big[ F_D(\bar{w}^S) \big] \leq \inf_{w \in \mathbb{R}^d} \Big\{ F_D(w) + \tilde{O}\Big( \frac{L(\|w\|^2 + 1)}{\sqrt{n}} \Big) \Big\}.$$

Thus, Theorem 3.1 shows that using GD, in the case of GLMs, we can learn the class $\mathcal{W}^B$, with sample complexity $m(\varepsilon) = \tilde{O}(1/\varepsilon^2)$ and number of iterations $T = \tilde{O}(1/\varepsilon^2)$.

Note that, an improved rate of $\tilde{O}(L\|w\|/\sqrt{n})$ can be obtained, if we choose $\eta = O(\|w\|/(L\sqrt{T}))$, but this requires prior knowledge of the bounded domain radius. The bound we formulate above is for a learning rate that is oblivious to the norm of the optimal choice $w$.

The key technical tool in proving Theorem 3.1 is the following structural result that characterizes the output of GD over the empirical risk, and may be of independent interest:

**Theorem 3.2.** *Let $S = \{z_1, \ldots, z_n\}$ be an i.i.d sequence drawn from some unknown distribution $D$. Assume that $f : \mathbb{R}^d \times \mathcal{Z} \to \mathbb{R}$ is a convex and $L$-Lipschitz function, for all $z \in \mathcal{Z}$, with respect to its first argument. Then, given the distribution $D$ there exists a sequence $u_1, \ldots, u_T$ that depends on $D$ (but independent of the sample $S$), such that, for any $\delta \in (0, 1)$ with probability of at least $1 - \delta$ over $S$, the iterates of gradient descent, as depicted in Eq. (2), satisfy:*

$$\forall t \in [T] : \quad \|w_t^S - u_t\| \leq O\Big( \frac{\eta L t}{\sqrt{n}} \sqrt{\log(T/\delta)} + \eta L \sqrt{t} \Big)$$

For example, we can set $\eta = \tilde{O}(1/\sqrt{T})$ and $T = O(n)$, then with probability $O(1/n)$:

$$\left\| w_t^S - u_t \right\| = O(1).$$

As such, for the natural choice of learning rate and iteration complexity, we obtain that the iterates of GD remain at the vicinity of a deterministic trajectory, predetermined by the distribution to be learned. Our following result shows that this bound is tight up to some logarithmic factor. We refer the reader to the supplementary material for the full proof.

**Theorem 3.3.** *Fix $\eta, L, T$ and $n$. For any sequence $u_1, \ldots, u_T$, independent of the sample $S$, there exists a convex and $L$-Lipschitz function $f : \mathcal{W} \times \mathcal{Z} \to \mathbb{R}$ and a distribution $D$ over $\mathcal{Z}$, such that, with probability at least $1/20$ over $S$, the iterates of gradient descent, as depicted in Eq. (2), satisfy:*

$$\forall t \in [T] : \quad \left\| w_t^S - u_t \right\| \geq \Omega\left(\frac{\eta L t}{\sqrt{n}} + \eta L \sqrt{t}\right).$$

## 3.1 High Probability Rates

We next move to discuss results for learnability with high probability. We first remark that using Markov's inequality and, standard techniques to boost the confidence, One can achieve high proability rates at a logarithmic computational and sample cost in the confidence.

If we want, though, to achieve high probability rates for the algorithm without alterations, the standard approach requires concentration bounds which normally rely on boundness of the predictor. This, unfortunately is not true when we run GD without projection, as the prediction can be potentially unbounded.

However, under natural structural assumptions that are often met for the types of losses we are usually interested in, we can achieve such boundness by *clipping* procedure which we next describe.

For this section we consider a loss function $\ell(a, y)$, such that $y \in [-b, b]$ and, we assume that:

$$\ell(a, y) \geq \begin{cases} \ell(|y|, y) & a \geq |y|, \\ \ell(-|y|, y) & a \leq -|y|. \end{cases} \quad \text{and} \quad \forall a \in [-b, b] : |\ell(a, y)| \leq c. \quad (9)$$

Note that this is a very natural assumption to have in the case of convex surrogates for prediction tasks. For example, this holds for the widely used hinge loss $\ell(a, y) = \max\{0, 1 - ya\}$ in binary classification. Observe that under this assumption, if we consider $w \cdot \phi(x)$ as a predictor of $y$, the learner has no incentive to return a prediction that is outside of the interval $[-b, b]$.

Thus, we define the following mapping $g(a)$

$$g(a) = \begin{cases} c & a \geq b, \\ a & a \in [-b, b], \\ -c & a \leq -b, \end{cases} \quad (10)$$

and consider the *clipped* solution $g(\bar{w}^S \cdot \phi(x))$ where $\bar{w}^S$ is the original output of GD defined in Eq. (3). We can now present our high probability result.

**Theorem 3.4.** *Suppose $D$ is a distribution over $\mathcal{Z} = \mathcal{X} \times \mathcal{Y}$ and let $\mathcal{W}^B = \{w \in \mathbb{R}^d : \|w\| \leq B\}$. Let $\ell(a, y)$ be a generalized linear loss function of the form given in Eqs. (6) and (9) such that for all $y \in \mathcal{Y}$, $\ell(\cdot, y)$ is convex and $L$-Lipschitz with respect to its first argument. Then, for the gradient descent solution in Eq. (3) and the function $g(a)$ in Eq. (10) we have with probability at least $1 - \delta$*

$$\mathop{\mathbb{E}}_{(x,y)\sim D}\left[\ell\big(g\big(\bar{w}^S \cdot \phi(x)\big), y\big)\right] \leq \inf_{w \in \mathbb{R}^d}\left\{\mathop{\mathbb{E}}_{(x,y)\sim D}\left[\ell\big(w \cdot \phi(x), y\big)\right] + \frac{\|w\|^2}{\eta T} + \|w\| L\sqrt{\frac{\log(2/\delta)}{n}}\right\}$$

$$+ O\left(\frac{\eta L^2 T}{n}\sqrt{\log(4T/\delta)} + \frac{\eta L^2 \sqrt{T}}{\sqrt{n}} + \eta L^2 + c\sqrt{\frac{\log(8/\delta)}{n}}\right).$$

In particular, setting $\eta = 1/(L\sqrt{T})$ and $T = n$ we obtain that with probability $1 - \delta$:

$$\mathop{\mathbb{E}}_{(x,y)\sim D}\left[\ell\big(g\big(\bar{w}^S \cdot \phi(x)\big), y\big)\right] \leq \inf_{w \in \mathcal{W}^1}\left\{\mathop{\mathbb{E}}_{(x,y)\sim D}\left[\ell\big(w \cdot \phi(x), y\big)\right]\right\} + \tilde{O}\left(\frac{L+c}{\sqrt{n}}\log(1/\delta)\right)$$

# 4 Comparison with non-distribution-dependent uniform convergence

In this section, we contrast our approach with what might be possible with a more traditional, non-distribution-dependent, uniform convergence. In particular, we consider an argument based on ensuring that the output of GD, with some stepsize $\eta$ and number of iterations $T$, is always (or with high probability) in a ball of radius $B(\eta, T)$ around the origin, namely, $\|\bar{w}^S\| \leq B(\eta, T)$. Hence, using Rademacher complexity bounds for GLMs, we can say that its population risk is within $O(BL/\sqrt{n})$ of its empirical risk. By selecting $\eta$ and $T$ to be very small, one can always ensure that the output of GD is in a small ball, but we also need to balance that with the empirical suboptimality of the output. That is, traditional bounds require us to find $\eta$ and $T$ that balance between the guarantee on the empirical suboptimality and the guarantee on the norm of the output. Such an approach would then result in population suboptimality that is governed by these two terms:

$$G(n) := \inf_{\eta, T} \sup_{D, f} \left\{ \max \left\{ \mathbb{E}_{S \sim D^n} \left[ F_S(\bar{w}^S) - \min_{w \in \mathcal{W}^1} F_S(w) \right], \mathbb{E}_{S \sim D^n} \left[ \frac{\|\bar{w}^S\| L}{\sqrt{n}} \right] \right\} \right\}, \tag{11}$$

where $D$ and $f$ are taken over all valid distributions such that $f$ is convex and Lipschitz. In particular, we would get,

$$\mathbb{E}_{S \sim D^n} \left[ F_D(\bar{w}^S) \right] \leq \min_{w \in \mathcal{W}^1} F_D(w) + O(G(n)).$$

**Claim 4.1.** *For GLMs it holds that $G(n) = \Theta(L/n^{1/4})$.*

**Proof.** The upper bound follows by taking $\eta = 1/(L\sqrt{T})$ and $T = \sqrt{n}$, bounding the first term using the standard GD guarantee in Eq. (4), and the second term by noting that each step increases the gradient by at most $\eta L$. Therefore, $\|\bar{w}^S\| \leq \eta L T$ and we obtain,

$$G(n) \leq \eta L^2 + \frac{1}{\eta T} + \frac{\eta L^2 T}{\sqrt{n}} = O\left( \frac{L}{n^{1/4}} \right).$$

For the lower bound we consider two deterministic functions in one dimension. When $\eta T \geq n^{1/4}/L$ we consider the deterministic objective $f(w; z) = Lw$. Clearly, the norm of the solution is $\|\bar{w}^S\| = \frac{1}{T} \sum_{t=1}^{T} \eta L t \geq \eta L T/2$, thus:

$$\frac{\|\bar{w}^S\| L}{\sqrt{n}} \geq \Omega\left( \frac{L}{n^{1/4}} \right).$$

When $\eta T < n^{1/4}/L$ we consider the objective $f(w; z) = Lw/n^{1/4}$. Similarly, we have that $\bar{w}^S = -\eta L(T+1)/(2n^{1/4})$. Thus, the empirical risk is:

$$F_S(\bar{w}^S) - \min_{w \in \mathcal{W}^1} F_S(w) = -\frac{\eta L^2(T+1)}{2n^{1/8}} + \frac{L}{n^{1/4}} \geq \frac{L}{2n^{1/4}} - \frac{L}{2Tn^{1/4}} \geq \Omega\left( \frac{L}{n^{1/4}} \right).$$

∎

The claim shows that uniform convergence to a fixed ball around the origin *can* ensure learning, but with a suboptimal sample complexity of $O(1/\varepsilon^4)$. To obtain better bounds using this approach, additional structural assumption on the loss or the distribution are required. For example, the work of [34] assumes max-margin and smoothness(specifically the logistic loss) to obtain bounds on the norm of the solution, while our result applies to general Lipschitz GLMs without any distribution assumptions. It is insightful to directly contrast Eq. (11) to the approach of Section 3, which essentially entails looking at:

$$\tilde{G}(n) := \inf_{\eta, T} \sup_{D, f} \left\{ \max \left\{ \mathbb{E}_{S \sim D^n} \left[ F_S(\bar{w}^S) - \min_{w \in \mathcal{W}^1} F_S(w) \right], \mathbb{E}_{S \sim D^n} \left[ \frac{\|\bar{w}^S - u\| L}{\sqrt{n}} \right] \right\} \right\},$$

for some deterministic $u$. We can then apply a uniform concentration guarantee for a ball around it, and so $\tilde{G}$ is also sufficient to ensure:

$$\mathbb{E}_{S \sim D^n} \left[ F_D(\bar{w}^S) \right] \leq \min_{w \in \mathcal{W}^1} F_D(w) + O(\tilde{G}(n)).$$

In Section 3 we show that $\tilde{G}(n) = \tilde{O}(L/\sqrt{n})$, improving on $G(n)$ and yielding optimal learning, up to logarithmic factors.

## 5 Technical Overview

Our main result, Theorem 3.1, establishes a generalization bound for GD without projection, and it builds upon a structural result presented in Theorem 3.2. We next, then, outline the derivation of both these results. Because the most interesting implication of our results are derived when we choose $\eta = \tilde{O}(1/\sqrt{T})$ and $T = O(n)$, we will focus here in the exposition on this regime. This is mainly to avoid clutter notation. Hence, unless stated otherwise we assume that $T$ and $\eta$ are fixed appropriately.

Our proof of Theorem 3.1 relies on two steps. First, through Theorem 3.2 we argue that that output of GD will be (w.h.p) in a fixed ball that depends solely on the distribution (but not on the sample). Then, as a second step, we can apply standard uniform convergence for a bounded-norm balls (see Lemma 2.1) to reason about generalization.

The second step, builds on a standard generalization bound that is derived through Rademacher complexity. Also notice, that the first step follows immediately from Theorem 3.2. Indeed, Theorem 3.2 argues that there exist a sequence $u_1, \ldots, u_T$ such that if $w_t^S$ is the trajectory of GD over the empirical risk, we will have (w.h.p):

$$\|w_t^S - u_t\| = O(1).$$

The output of GD is the averaged iterate, hence we obtain that indeed $\bar{w}^S$ is restricted to a ball around $u = \frac{1}{T} \sum_{t=1}^{T} u_t$, as required. Therefore, we are left with proving our main structural result: That the iterates of GD are in a proximity of a trajectory $u_1, \ldots, u_T$ that depends solely on the distribution (i.e. Theorem 3.2).

For the end of proving Theorem 3.2, we introduce the following GD trajectory:

**Gradient Descent on the population loss.** We consider an alternative sequence of gradient descent that operates on the population loss $F_D(w)$ rather than the empirical risk $F_S(w)$. The update rule is then,

$$w_{t+1}^D = w_t^D - \eta \nabla F_D(w_t^D). \tag{12}$$

The sequence $w_1^D, \ldots, w_t^D$ will serve as the sequence $u_1, \ldots, u_T$.

In the proof of Theorem 3.2 we require high probability rates. However, for the simplicity of the exposition, we will prove here a slightly weaker, in expectation, result: (the proof is defered to the supplementary material).

**Lemma 5.1.** *Let $S = \{z_1, \ldots, z_n\}$ be an i.i.d sequence drawn from some unknown distribution $D$. Assume that $f : \mathbb{R}^d \times \mathcal{Z} \to \mathbb{R}$ is a convex and $L$-Lipschitz function, for all $z \in \mathcal{Z}$, with respect to its first argument. Then, the iterates of gradient descent, as depicted in Eqs. (2) and (12), satisfy:*

$$\forall t \in [T] : \quad \mathbb{E}_{S \sim D^n} \left[ \|w_t^S - w_t^D\| \right] \leq O\left( \frac{\eta L t}{\sqrt{n}} + \eta L \sqrt{t} \right)$$

This lemma bounds the distance, in expectation, between the GD trajectory over the empirical risk and the GD trajectory over the population risk. Again, we remark that to obtain the final result in Theorem 3.2, a stronger high probability version of Lemma 5.1 is required. The proof of Theorem 3.2 follows similar lines to that of Lemma 5.1 with modifications concerning specific concentration inequalities.

One crucial challenge in proving Lemma 5.1 stems from the adaptivity of the gradient sequence. In particular, notice that the sequences $w_t^S, w_t^D$ are governed, respectively, by the dynamics

$$w_t^S = w_{t-1}^S - \eta \nabla F_S(w_t^S), \quad w_t^D = w_{t-1}^D - \eta \nabla F_D(w_t^D).$$

At a first glance, since $\nabla F_S(w)$ is an estimate of $\nabla F_D(w)$, it might seem that the result can be obtained by standard concentration bounds and application of a union bound along the trajectory. Unfortunately, such a naive argument cannot work. Indeed, since $w_t$, for $t \geq 2$, depends on the sequence $S$, $\nabla F_S(w_t)$ is not necessarily an unbiased estimate of $\nabla F_D(w_t)$. This is not just seemingly, and a construction in [2] demonstrates how, even after only two iterates the gradient $\nabla F_S(w_t)$ can diverge, significantly, from $\nabla F_D(w_t)$. While these constructions are outside of the scope of GLMs,

we remark that Theorem 3.2 holds for *any* convex and Lipschitz function. Also, it was in fact shown that even for GLMs [11], the estimate of the gradients don't have any dimension-independent uniform convergence bound, making it a challenge to pursue such a proof direction. To summarize, we need to prove that the two sequences remain in the vicinity, even though, apriori, the update steps of the two functions may be different at each iteration.

Towards this goal, we follow an analysis, reminiscent of the uniform argument stability analysis of [5] for non-smooth convex losses. In a nutshell, both arguments compare the trajectory to a reference trajectory, and bound the incremental difference between the two trajectories while exploiting monotonicity of the convex function. In the case of stability, the reference trajectory is the trajectory $w_t^{S'}$ induced by a sample $S' = (z_1, \ldots, z_i', \ldots, z_n)$ that differ on a single example from the sample $S = (z_1, \ldots, z_i, \ldots, z_n)$.

Bassily et al. showed that if $S$ and $S'$ are two sequences that differ on a single example, then:

$$\|w_t^S - w_t^{S'}\| = O\left(\frac{\eta T}{n} + \eta\sqrt{T}\right).$$

In comparison, we show:

$$\mathop{\mathbb{E}}_{S \sim D^n}\left[\|w_t^S - w_t^D\|\right] = O\left(\frac{\eta T}{\sqrt{n}} + \eta\sqrt{T}\right).$$

We note, that both results are optimal. Namely, the stability bound of [5] is the optimal stability rate, and the proximity bound we provide is the best possible bound against a fixed point, independent of the sample.

Note that both bounds yield $O(1)$ difference for the natural choice of $\eta$ and $T$. However, $O(1)$-stability guarantees are vacuous in the sense that they do not provide any interesting implication over the generalization of the algorithm. Whereas, we provide $O(1)$-proximity guarantee to some fixed point, completely independent of the sample $S$. This does imply generalization in our setup.

One might also suggest that our result of $O(1)$-proximity can be derived from $O(1)$-stability. However, it turns out this is not the case. For the same instance we use to lower bound the proximity by $\Omega(\eta T/\sqrt{n})$ in Theorem 3.3, it can be easily shown that GD will be $O(\eta T/n)$-stable. Setting $\eta = O(1)$ and $T = O(n)$ will then imply $O(1)$-stability but with $\Omega(\sqrt{n})$-proximity. This asserts that our result is not a direct consequence of standard stability arguments.

**Acknowledgements** RL received support from the Israeli Science Foundation (ISF) (grant no 2188/20) and from a Google research scholar award. The work was funded in part by the Israel Council for Higher Education Data-Science Centers.

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
