# Supplementary Material

## A    Main Proofs

Recall that we define $w_t^D$ to be the $t$-th iterate when applying GD over the population risk as depicted in Eq. (12).

### A.1    Proof of Lemma 5.1

Observe that,

$$
\begin{aligned}
\|w_{t+1}^S - w_{t+1}^D\|^2 &= \|w_t^S - w_t^D - \eta(\nabla F_S(w_t^S) - \nabla F_D(w_t^D))\|^2 \\
&= \|w_t^S - w_t^D\|^2 - 2\eta(\nabla F_S(w_t^S) - \nabla F_D(w_t^D))(w_t^S - w_t^D) + \eta^2\|\nabla F_S(w_t^S) - \nabla F_D(u_t)\|^2 \\
&\leq \|w_t^S - w_t^D\|^2 - 2\eta(\nabla F_S(w_t^S) - \nabla F_D(w_t^D))(w_t^S - w_t^D) + 4\eta^2 L^2 \quad (L\text{-Lipschitz}) \\
&= \|w_t^S - w_t^D\|^2 - 2\eta(\nabla F_S(w_t^S) - \nabla F_S(w_t^D))(w_t^S - w_t^D) \\
&\qquad\qquad + 2\eta(\nabla F_D(w_t^D) - \nabla F_S(w_t^D))(w_t^S - w_t^D) + 4\eta^2 L^2 \\
&\leq \|w_t^S - w_t^D\|^2 + 2\eta(\nabla F_D(w_t^D) - \nabla F_S(w_t^D))(w_t^S - w_t^D) + 4\eta^2 L^2,
\end{aligned}
$$

where in the last inequality we use monotinicity of convex functions: $(\nabla F_S(w) - \nabla F_S(u))(w - u) \geq 0$ for any $w, u$. Next, applying Cauchy-Schwarz inequality we get,

$$
\begin{aligned}
\|w_{t+1}^S - w_{t+1}^D\|^2 &\leq \|w_t^S - w_t^D\|^2 + 2\eta(\nabla F_D(w_t^D) - \nabla F_S(w_t^D))(w_t^S - w_t^D) + 4\eta^2 L^2 \\
&\leq \|w_t^S - w_t^D\|^2 + 2\eta\|\nabla F_S(w_t^D) - \nabla F_D(w_t^D)\|\|w_t^S - w_t^D\| + 4\eta^2 L^2 \quad (\text{C.S.}) \\
&\leq \|w_t^S - w_t^D\|^2 + \eta^2 t\|\nabla F_S(w_t^D) - \nabla F_D(w_t^D)\|^2 + \frac{1}{t}\|w_t^S - w_t^D\|^2 + 4\eta^2 L^2 \\
&= \left(1 + \frac{1}{c}\right)\|w_t^S - w_t^D\|^2 + \eta^2 c\|\nabla F_S(w_t^D) - \nabla F_D(w_t^D)\|^2 + 4\eta^2 L^2.
\end{aligned}
$$

The last inequality follows from the observation that $2ab \leq ca^2 + b^2/c$ for any $c > 0$ and $a, b \geq 0$, specifically for $a = \eta\|\nabla F_S(w_t^D) - \nabla F_D(w_t^D)\|$, and $b = \|w_t^S - w_t^D\|$.

Applying the formula recursively, and noting that $w_0 = u_0$:

$$
\begin{aligned}
\|w_{t+1}^S - w_{t+1}^D\|^2 &\leq \sum_{t'=0}^{t}\left(1 + \frac{1}{c}\right)^{t-t'}\left(\eta^2 c\|\nabla F_S(w_{t'}^D) - \nabla F_D(w_{t'}^D)\|^2 + 4\eta^2 L^2\right) \\
&\leq \sum_{t'=0}^{t}\left(e\eta^2 t\|\nabla F_S(w_{t'}^D) - \nabla F_D(w_{t'}^D)\|^2 + 4e\eta^2 L^2\right),
\end{aligned}
$$

where in the last inequality we chose $c = t + 1$ and used the known bound of $(1 + 1/t)^t \leq e$. Taking the square root and using the inequality of $\sqrt{a + b} \leq \sqrt{a} + \sqrt{b}$ we conclude

$$
\|w_{t+1}^S - w_{t+1}^D\| \leq \sqrt{e\eta^2(t + 1)\sum_{t'=0}^{t}\|\nabla F_S(w_{t'}^D) - \nabla F_D(w_{t'}^D)\|^2} + 2\sqrt{e}\eta L\sqrt{t + 1}.
$$

We are interested in bounding $\mathbb{E}_{S\sim D^n}\left[\|\nabla F_S(w_t^D) - \nabla F_D(w_t^D)\|^2\right]$. By the definition of the empirical risk

$$
\underset{S\sim D^n}{\mathbb{E}}\left[\|\nabla F_S(w_t^D) - \nabla F_D(w_t^D)\|^2\right] = \frac{1}{n^2}\underset{S\sim D^n}{\mathbb{E}}\left[\left\|\sum_{i=1}^{n}\left[\nabla f(w_t^D; z_i) - \underset{z\sim D}{\mathbb{E}}[\nabla f(w_t^D; z)]\right]\right\|^2\right]
$$

Note that by Lipschitzness $\|\nabla f(w_t^D; z_i) - \mathbb{E}[\nabla f(w_t^D; z)]\| \leq 2L$, and that $\{\nabla f(w_t^D; z_i) - \mathbb{E}[\nabla f(w_t^D; z)]\}_{i\in[n]}$ are independent zero-mean random vectors (as $w_t^D$ is independent of $z_i$). Thus, we get

$$
\underset{S\sim D^n}{\mathbb{E}}\left[\|\nabla F_S(w_t^D) - \nabla F_D(w_t^D)\|^2\right] = \frac{1}{n^2}\sum_{i=1}^{n}\left[\underset{S\sim D^n}{\mathbb{E}}\left[\|\nabla f(w_t^D; z_i) - \underset{z\sim D}{\mathbb{E}}[\nabla f(w_t^D; z)]\|^2\right]\right] \leq \frac{4L^2}{n}.
$$

Consequently, taking the expectation over the sample $S$ and using Jensen's inequality

$$\mathop{\mathbb{E}}_{S \sim D^n}\left[\|w_{t+1}^S - w_{t+1}^D\|\right] \leq \frac{4\eta L(t+1)}{\sqrt{n}} + 4\eta L\sqrt{t+1}.$$

## A.2 Proof of Theorem 3.1

Starting with Lemma 2.1 we obtain for any domain $\mathcal{W}_u^K$,

$$\mathop{\mathbb{E}}_{S \sim D^n}\left[\sup_{w \in \mathcal{W}_u^K} \{F_D(w) - F_S(w)\}\right] \leq \frac{2LK}{\sqrt{n}}. \tag{13}$$

From Theorem 3.2, there exists a sequence $u_1, \ldots, u_T$ such that with probability at least $1 - \delta$,

$$\left\|\bar{w}^S - \frac{1}{T}\sum_{t=1}^{T} u_t\right\| \leq \frac{7\eta LT}{\sqrt{n}}\sqrt{\log(T/\delta)} + 4\eta L\sqrt{T}. \tag{14}$$

Setting $u = \frac{1}{T}\sum_{t=1}^{T} u_t$ and $K$ to be the RHS in Eq. (14) we obtain:

$$\begin{aligned}
\mathop{\mathbb{E}}_{S \sim D^n}\left[F_D(\bar{w}^S) - F_S(\bar{w}^S)\right] &\leq \mathop{\mathbb{E}}_{S \sim D^n}\left[\sup_{w \in \mathcal{W}_u^K} \{F_D(w) - F_S(w)\}\big|\bar{w}^S \in \mathcal{W}_u^K\right] P(\bar{w}^S \in \mathcal{W}_u^K)\\
&\quad + P(\bar{w}^S \notin \mathcal{W}_u^K) \sup_{S \sim D^n} |F_D(\bar{w}^S) - F_S(\bar{w}^S)|\\
&= \mathop{\mathbb{E}}_{S \sim D^n}\left[\sup_{w \in \mathcal{W}_u^K} \{F_D(w) - F_S(w)\}\right] - \mathop{\mathbb{E}}_{S \sim D^n}\left[\sup_{w \in \mathcal{W}_u^K} \{F_D(w) - F_S(w)\}\big|\bar{w}^S \notin \mathcal{W}_u^K\right] P(\bar{w}^S \notin \mathcal{W}_u^K)\\
&\quad + P(\bar{w}^S \notin \mathcal{W}_u^K) \sup_{S \sim D^n} |F_D(\bar{w}^S) - F_S(\bar{w}^S)|\\
&\leq \mathop{\mathbb{E}}_{S \sim D^n}\left[\sup_{w \in \mathcal{W}_u^K} \{F_D(w) - F_S(w)\}\right] + P(\bar{w}^S \notin \mathcal{W}_u^K) \sup_{S \sim D^n} \sup_{w \in \mathcal{W}_u^K} |F_D(w) - F_S(w)|\\
&\quad + P(\bar{w}^S \notin \mathcal{W}_u^K) \sup_{S \sim D^n} |F_D(\bar{w}^S) - F_S(\bar{w}^S)|\\
&\leq \frac{14\eta L^2 T}{n}\sqrt{\log(T/\delta)} + \frac{8\eta L^2 \sqrt{T}}{\sqrt{n}} + P(\bar{w}^S \notin \mathcal{W}_u^K) \sup_{S \sim D^n} \sup_{w \in \mathcal{W}_u^K} |F_D(w) - F_S(w)|\\
&\quad + P(\bar{w}^S \notin \mathcal{W}_u^K) \sup_{S \sim D^n} |F_D(\bar{w}^S) - F_S(\bar{w}^S)| \qquad \text{(Eqs. (13) and (14))}\\
&\leq \frac{14\eta L^2 T}{n}\sqrt{\log(T/\delta)} + \frac{8\eta L^2 \sqrt{T}}{\sqrt{n}} + O\left(\delta\eta L^2 T\sqrt{\log(T/\delta)}\right),
\end{aligned}$$

where we used Eq. (7) and the fact that $\|\bar{w}^S\| \leq \eta LT$ and $\|u\| + K \leq O\left(\eta LT + \eta LT\sqrt{\log(T/\delta)}/\sqrt{n}\right) \leq O\left(\eta LT\sqrt{\log(T/\delta)}\right)$ to bound the second and third terms. Hence:

$$|F_D(\bar{w}^S) - F_S(\bar{w}^S)| \leq |F_D(\bar{w}^S)| + |F_S(\bar{w}^S)| \leq O\left(\eta L^2 T\right),$$

and

$$\sup_{w \in \mathcal{W}_u^K} |F_D(w) - F_S(w)| \leq \sup_{w \in \mathcal{W}_u^K} |F_D(w)| + \sup_{w \in \mathcal{W}_u^K} |F_S(w)| \leq O\left(\eta L^2 T\sqrt{\log(T/\delta)}\right).$$

Next, setting $\delta = O(1/\sqrt{nT})$ we get that:

$$\mathop{\mathbb{E}}_{S \sim D^n}\left[F_D(\bar{w}^S) - F_S(\bar{w}^S)\right] \leq O\left(\frac{\eta L^2 T}{n}\sqrt{\log(nT)} + \frac{\eta L^2 \sqrt{T}}{\sqrt{n}}\sqrt{\log(nT)}\right). \tag{15}$$

Finally, combining Eqs. (4) and (15) we obtain that for every $w^\star \in \mathcal{W}^B$:

$$\begin{aligned}
\mathop{\mathbb{E}}_{S \sim D^n}\left[F_D(\bar{w}^S)\right] - F_D(w^\star) &= \mathop{\mathbb{E}}_{S \sim D^n}\left[F_D(\bar{w}^S) - F_S(\bar{w}^S)\right] + \mathop{\mathbb{E}}_{S \sim D^n}\left[F_S(\bar{w}^S)\right] - F_D(w^\star)\\
&= \mathop{\mathbb{E}}_{S \sim D^n}\left[F_D(\bar{w}^S) - F_S(\bar{w}^S)\right] + \mathop{\mathbb{E}}_{S \sim D^n}\left[F_S(\bar{w}^S) - F_S(w^\star)\right]\\
&\leq O\left(\frac{\eta L^2 T}{n}\sqrt{\log(nT)} + \frac{\eta L^2 \sqrt{T}}{\sqrt{n}}\sqrt{\log(nT)} + \eta L^2 + \frac{B^2}{\eta T}\right).
\end{aligned}$$

## A.3 Proof of Theorem 3.2

The proof is similar to that of Lemma 5.1 with the exception that here we employ specific concentration inequalities of random variables with bounded difference. The reference sequence we consider is the GD iterates over the population risk, namely, $w_t^D$ as described in Eq. (12). Observe that

$$
\begin{aligned}
\|w_{t+1}^S - w_{t+1}^D\|^2 &= \|w_t^S - w_t^D - \eta(\nabla F_S(w_t^S) - \nabla F_D(w_t^D))\|^2 \\
&= \|w_t^S - w_t^D\|^2 - 2\eta(\nabla F_S(w_t^S) - \nabla F_D(w_t^D))(w_t^S - w_t^D) + \eta^2\|\nabla F_S(w_t^S) - \nabla F_D(w_t^D)\|^2 \\
&\leq \|w_t^S - w_t^D\|^2 - 2\eta(\nabla F_S(w_t^S) - \nabla F_D(w_t^D))(w_t^S - w_t^D) + 4\eta^2 L^2 \quad (L\text{-Lipschitz}) \\
&= \|w_t^S - w_t^D\|^2 - 2\eta(\nabla F_S(w_t^S) - \nabla F_S(w_t^D))(w_t^S - w_t^D) \\
&\qquad + 2\eta(\nabla F_D(w_t^D) - \nabla F_S(w_t^D))(w_t^S - w_t^D) + 4\eta^2 L^2.
\end{aligned}
$$

From convexity of $F_S$ we know that $(\nabla F_S(w) - \nabla F_S(u))(w - u) \geq 0$ for any $w, u$. Therefore, applying Cauchy-Schwarz inequality we get,

$$
\begin{aligned}
\|w_{t+1}^S - w_{t+1}^D\|^2 &\leq \|w_t^S - w_t^D\|^2 + 2\eta(\nabla F_D(w_t^D) - \nabla F_S(w_t^D))(w_t^S - w_t^D) + 4\eta^2 L^2 \quad \text{(convexity)} \\
&\leq \|w_t^S - w_t^D\|^2 + 2\eta\|\nabla F_S(w_t^D) - \nabla F_D(w_t^D)\|\|w_t^S - w_t^D\| + 4\eta^2 L^2 \quad \text{(C.S.)} \\
&\leq \|w_t^S - w_t^D\|^2 + \eta^2 t\|\nabla F_S(w_t^D) - \nabla F_D(w_t^D)\|^2 + \frac{1}{t}\|w_t^S - w_t^D\|^2 + 4\eta^2 L^2 \\
&= \left(1 + \frac{1}{c}\right)\|w_t^S - w_t^D\|^2 + \eta^2 c\|\nabla F_S(w_t^D) - \nabla F_D(w_t^D)\|^2 + 4\eta^2 L^2. \quad (16)
\end{aligned}
$$

The last inequality follows from the observation that $2ab \leq ca^2 + b^2/c$ for any $c > 0$ and $a, b \geq 0$, specifically for $a = \eta\|\nabla F_S(w_t^D) - \nabla F_D(w_t^D)\|$, and $b = \|w_t^S - w_t^D\|$.

We are interested in bounding $\|\nabla F_S(w_t^D) - \nabla F_D(w_t^D)\|$. For that matter we consider the following concentration inequality which is a direct result of the bounded difference inequality by McDiarmid.

**Theorem** (Boucheron, Lugosi, and Massart [7, Example 6.3]). *Let $X_1, \ldots, X_n$ be independent zero-mean R.V's such that $\|X_i\| \leq c_i/2$ and denote $v = \frac{1}{4}\sum_{i=1}^n c_i^2$. Then, for all $t \geq \sqrt{v}$,*

$$
\mathbb{P}\left\{\left\|\sum_{i=1}^n X_i\right\| > t\right\} \leq e^{-(t-\sqrt{v})^2/(2v)}.
$$

Note that by Lipschitzness $\|\nabla f(w_t^D; z_i) - \mathbb{E}[\nabla f(w_t^D; z)]\| \leq 2L$, and that $\{\nabla f(w_t^D; z_i) - \mathbb{E}[\nabla f(w_t^D; z)]\}_{i \in [n]}$ are independent zero-mean random variables (as $w_t^D$ is independent of $z_i$). Thus, for $\Delta \geq 2\frac{L}{\sqrt{n}}$:

$$
\mathbb{P}\left\{\left\|\frac{1}{n}\sum_{i=1}^n \nabla f(w_t^D; z_i) - \mathbb{E}[\nabla f(w_t^D; z)]\right\| > \Delta\right\} \leq e^{-(\Delta\sqrt{n} - 2L)^2/(4L^2)},
$$

This implies that with probability $1 - \delta$ we get,

$$
\|\nabla F_S(w_t^D) - \nabla F_D(w_t^D)\| \leq \frac{4L}{\sqrt{n}}\sqrt{\log(1/\delta)}.
$$

Plugging it back to Eq. (16) we obtain w.p. $1 - \delta$

$$
\|w_{t+1}^S - w_{t+1}^D\|^2 \leq \left(1 + \frac{1}{c}\right)\|w_t^S - w_t^D\|^2 + \frac{16\eta^2 L^2 c \log(1/\delta)}{n} + 4\eta^2 L^2
$$

Applying the formula recursively, and noting that $w_0^S = w_0^D$:

$$
\begin{aligned}
\|w_{t+1}^S - w_{t+1}^D\|^2 &\leq \sum_{t'=0}^t \left(1 + \frac{1}{c}\right)^{t'}\left(\frac{16\eta^2 L^2 c \log(1/\delta)}{n} + 4\eta^2 L^2\right) \\
&\leq \frac{16e\eta^2 L^2 (t+1)^2 \log(1/\delta)}{n} + 4e\eta^2 L^2(t+1),
\end{aligned}
$$

where in the last inequality we chose $c = t + 1$ and used the known bound of $(1 + 1/t)^t \leq e$. Taking the square root and using the inequality of $\sqrt{a+b} \leq \sqrt{a} + \sqrt{b}$ we have

$$\|w_{t+1}^S - w_{t+1}^D\| \leq 7\sqrt{\frac{\eta^2 L^2 (t+1)^2 \log(1/\delta)}{n}} + 4\eta L \sqrt{t+1}.$$

By taking the union bound over all $t \in [T]$ we conclude the proof.

## A.4 Proof of Theorem 3.4

Similarly to the proof in the supplementary material, let us consider the domain $\mathcal{W}_u^K = \{w : \|w - u\| \leq K\}$, where we set $u = \frac{1}{T} \sum_{t=1}^T u_t$, the average of the deterministic sequence in Theorem 3.2. From the assumption in Eq. (9) it follows that for any $w$ we have that $|\ell(g(w \cdot \phi(x)), y)| \leq c$. We also, can use Lemma 2.1 (applying it to $\ell \circ g \to \ell$) to obtain that

$$\mathbb{E}_{S \sim D^n} \left[ \sup_{w \in \mathcal{W}_u^K} \{F_D(g \circ w) - F_S(g \circ w)\} \right] \leq \frac{2LK}{\sqrt{n}}, \tag{17}$$

where we denote

$$F_S(g \circ w) = \frac{1}{n} \sum_{i=1}^n \ell(g(w \cdot \phi(x_i), y_i)), \quad F_D(g \circ w) = \mathbb{E}_{(x,y) \sim D} [\ell(g(w \cdot \phi(x), y))].$$

Next, we define

$$G(S) = \sup_{w \in \mathcal{W}_u^k} \{F_D(g \circ w) - F_S(g \circ w)\},$$

and note that for two samples, $S, S'$ that differ on a single example we have that

$$|G(S) - G(S')| \leq \frac{2c}{n}.$$

Using then the bounded difference inequality by McDiarmid [see 33, Lemma 26.4]. We have that with probability at least $1 - \delta$,

$$G(S) = \sup_{w \in \mathcal{W}_u^K} \left\{ \mathbb{E}_{(x,y) \sim D} [\ell(g(w \cdot \phi(x)), y)] - \frac{1}{n} \sum_{i=1}^n \ell(g(w \cdot \phi(x_i)), y_i) \right\}$$

$$\leq \mathbb{E}_{S \sim D^n} [G(S)] + c\sqrt{\frac{2 \log(2/\delta)}{n}} \qquad \text{(McDiarmid)}$$

$$\leq \frac{2LK}{\sqrt{n}} + c\sqrt{\frac{2 \log(2/\delta)}{n}}.$$

From Theorem 3.2 we have that with probability at least $1 - \delta$,

$$\|\bar{w}^S - u\| \leq \frac{6\eta LT}{\sqrt{n}} \sqrt{\log(T/\delta)} + 4\eta L \sqrt{T} \tag{18}$$

Taken together, and applying union bound, we have that with probability at least $1 - \delta$:

$$\mathbb{E}_{(x,y) \sim D} [\ell(g(\bar{w}^S \cdot \phi(x)), y)] \leq \frac{1}{n} \sum_{i=1}^n \ell(g(\bar{w}^S \cdot \phi(x_i)), y_i)$$

$$+ \frac{12\eta L^2 T}{n} \sqrt{\log(2T/\delta)} + \frac{8\eta L^2 \sqrt{T}}{\sqrt{n}} + c\sqrt{\frac{2 \log(4/\delta)}{n}}. \tag{19}$$

Next, using Eqs. (4) and (9) and the fact that the optimization bound Eq. (4) holds for any $B > 0$:

$$\frac{1}{n}\sum_{i=1}^{n}\ell\big(g\big(\bar{w}^S\cdot\phi(x_i)\big),y_i\big) \leq \frac{1}{n}\sum_{i=1}^{n}\ell\big(\bar{w}^S\cdot\phi(x_i),y_i\big) \qquad\qquad\text{(Eq. (9))}$$

$$\leq \inf_{B\in\mathbb{R}^+}\left\{\min_{w\in\mathcal{W}^B}\left\{\frac{1}{n}\sum_{i=1}^{n}\ell\big(w\cdot\phi(x_i),y_i\big)\right\} + \eta L^2 + \frac{B^2}{\eta T}\right\} \qquad\text{(Eq. (4))}$$

$$\leq \inf_{w\in\mathbb{R}^d}\left\{\frac{1}{n}\sum_{i=1}^{n}\ell\big(w\cdot\phi(x_i),y_i\big) + \eta L^2 + \frac{\|w\|^2}{\eta T}\right\}, \qquad\qquad (20)$$

Now, set $w^\star$ such that

$$\mathop{\mathbb{E}}_{(x,y)\sim D}\big[\ell\big(w^\star\cdot\phi(x),y\big)\big] + \frac{\|w^\star\|^2}{\eta T} + \|w^\star\|L\sqrt{\frac{2\log(1/\delta)}{n}}$$

$$\leq \inf_{w\in\mathbb{R}^d}\left\{\mathop{\mathbb{E}}_{(x,y)\sim D}\big[\ell\big(w\cdot\phi(x),y\big)\big] + \frac{\|w\|^2}{\eta T} + \|w\|L\sqrt{\frac{2\log(1/\delta)}{n}}\right\} + \eta L^2. \qquad (21)$$

By independence of $\{(x_i,y_i)\}_{i=1}^{n}$ and the bound on $|\ell(0,y)| \leq c$ we obtain by Lipschitzness $|\ell(w^\star\cdot\phi(x),y)| \leq \|w^\star\|L + c$. It follows from the Hoeffding's inequality that with probability at least $1-\delta$

$$\frac{1}{n}\sum_{i=1}^{n}\ell\big(w^\star\cdot\phi(x_i),y_i\big) - \mathop{\mathbb{E}}_{(x,y)\sim D}\big[\ell\big(w^\star\cdot\phi(x),y\big)\big] \leq (\|w^\star\|L + c)\sqrt{\frac{2\log(1/\delta)}{n}}. \qquad (22)$$

Thus, we have that w.p. $1-\delta$:

$$\frac{1}{n}\sum_{i=1}^{n}\ell\big(g\big(\bar{w}^S\cdot\phi(x_i)\big),y_i\big) \leq \inf_{w\in\mathbb{R}^d}\left\{\frac{1}{n}\sum_{i=1}^{n}\ell\big(w\cdot\phi(x_i),y_i\big) + \eta L^2 + \frac{\|w\|^2}{\eta T}\right\}$$

$$\leq \frac{1}{n}\sum_{i=1}^{n}\ell\big(w^\star\cdot\phi(x_i),y_i\big) + \eta L^2 + \frac{\|w^\star\|^2}{\eta T}$$

$$\leq \mathop{\mathbb{E}}_{(x,y)\sim D}\big[\ell\big(w^\star\cdot\phi(x),y\big)\big] + \frac{\|w^\star\|^2}{\eta T} + (\|w^\star\|L + c)\sqrt{\frac{2\log(1/\delta)}{n}} + \eta L^2$$

$$\text{(Eq. (22))}$$

$$\leq \inf_{w\in\mathbb{R}^d}\left\{\mathop{\mathbb{E}}_{(x,y)\sim D}\big[\ell\big(w\cdot\phi(x),y\big)\big] + \frac{\|w\|^2}{\eta T} + (\|w\|L + c)\sqrt{\frac{2\log(1/\delta)}{n}}\right\} + 2\eta L^2,$$

$$(23)$$

where the last inequality follows from Eq. (21). Combining Eqs. (19) and (23) and applying union bound we obtain the result.

## B  Proof of Lemma 2.1

Using the standard bound of the generalization error, via the Rademacher complexity of the class (see e.g. [33]), we have that:

$$\mathop{\mathbb{E}}_{S\sim D^n}\left[\sup_{w\in\mathcal{W}_u^K}\big\{F_D(w) - F_S(w)\big\}\right] \leq 2\mathop{\mathbb{E}}_{S\sim D^n}[\mathcal{R}_S(f\circ\mathcal{W}_u^K)],$$

Where we notate the function class:

$$f\circ\mathcal{W}_u^K = \{z \to \ell(w\cdot x,y) : w\in\mathcal{W}_u^K\}.$$

and $\mathcal{R}_S(f\circ\mathcal{W}_u^K)$ is the Rademacher complexity of the class $f\circ\mathcal{W}_u^K$. Namely:

$$\mathcal{R}_S(f\circ\mathcal{W}_u^K) := \mathop{\mathbb{E}}_{\sigma}\left[\sup_{h\in f\circ W_u^K}\frac{1}{n}\sum_{z_i\in S}\sigma_i h(z_i)\right], \qquad (24)$$

and $\sigma_1, \ldots, \sigma_n$ are i.i.d. Rademacher random variables.

We next show that:

$$\mathcal{R}_S(f \circ \mathcal{W}_u^K) \leq \frac{LK}{\sqrt{n}}. \tag{25}$$

To show Eq. (25), we use the following well known property of the Rademacher complexity of a class:

**Lemma B.1** (contraction lemma, see [33])**.** *For each* $i \in [n]$, *let* $\rho_i : \mathbb{R} \to \mathbb{R}$ *be convex L-lipschitz function in their first argument. Let* $A \subseteq \mathbb{R}^n$ *and denote* $a = (a_1, \ldots, a_n) \in A$. *Then, if* $\sigma = \sigma_1, \ldots, \sigma_n$, *are i.i.d. Rademacher random variables*

$$\mathbb{E}_{\sigma}\left[\sup_{a \in A} \sum_{i=1}^{n} \sigma_i \rho_i(a_i)\right] \leq L \cdot \mathbb{E}_{\sigma}\left[\sup_{a \in A} \sum_{i=1}^{n} \sigma_i a_i\right].$$

As well as the Rademacher complexity of the class of linear predictors against a sample $S = \{\phi(x_1), \ldots, \phi(x_n)\}$ of $\ell_2$ 1-bounded vectors:

$$\mathcal{R}_S(f \circ \mathcal{W}_0^K) = K/\sqrt{n}. \tag{26}$$

Next, given a sample $S = \{z_1, \ldots, z_n\}$ we define $\rho_i(\alpha) := \ell(\alpha + u \cdot \phi(x_i), y_i)$ and we set

$$A := \{(v \cdot \phi(x_1), \ldots, v \cdot \phi(x_n)) : v \in \mathcal{W}_0^k\}.$$

Then:

$$
\begin{aligned}
n\mathcal{R}_S(f \circ \mathcal{W}_u^K) &= \mathbb{E}_{\sigma}\left[\sup_{w \in \mathcal{W}_u^k} \sum_{i=1}^{n} \sigma_i \ell(w \cdot \phi(x_i), y_i)\right] \\
&= \mathbb{E}_{\sigma}\left[\sup_{w \in \mathcal{W}_u^k} \sum_{i=1}^{n} \sigma_i \ell((w-u) \cdot \phi(x_i) + u \cdot \phi(x_i), y_i)\right] \\
&= \mathbb{E}_{\sigma}\left[\sup_{v \in \mathcal{W}_0^k} \sum_{i=1}^{n} \sigma_i \ell(v \cdot \phi(x_i) + u \cdot \phi(x_i), y_i)\right] \\
&\leq L \cdot \mathbb{E}_{\sigma}\left[\sup_{v \in \mathcal{W}_0^k} \sum_{i=1}^{n} \sigma_i v \cdot \phi(x_i)\right] \qquad \text{(Lemma B.1)} \\
&\leq LK\sqrt{n} \qquad\qquad\qquad\qquad\qquad\quad \text{(Eq. (26))}
\end{aligned}
$$

Dividing by $n$ yields the proof.

## C  Proof of Theorem 3.3

Our construction is comprised of two separate instances. We first provide lower bounds, Lemmas C.1 and C.2, for the distance between the GD iterates $w_t^S, w_t^{S'}$ defined over two separate i.i.d. samples $S = (z_1, \ldots, z_n)$ and $S' = (z_1', \ldots, z_n')$, respectively.

**Lemma C.1.** *Fix* $\eta, L, T$ *and* $n$. *Suppose* $S$ *and* $S'$ *are i.i.d. samples drawn from* $D^n$. *There exists a convex and L-Lipschitz function* $f : \mathcal{W} \times \mathcal{Z} \to \mathbb{R}$ *and a distribution* $D$ *over* $\mathcal{Z}$, *such that, if* $w_t^S$ *and* $w_t^{S'}$ *are defined as in Eq.* (2)*, then with probability at least* $1/10$:

$$\forall t \in [T] : \quad \|w_t^S - w_t^{S'}\| \geq \Omega\left(\frac{\eta L t}{\sqrt{n}}\right).$$

**Lemma C.2.** *Fix* $\eta, L, T$ *and* $n$. *Suppose* $S$ *and* $S'$ *are i.i.d. samples drawn from* $D^n$. *There exists a convex and L-Lipschitz function* $f : \mathcal{W} \times \mathcal{Z} \to \mathbb{R}$ *and a distribution* $D$ *over* $\mathcal{Z}$, *such that, if* $w_t^S$ *and* $w_t^{S'}$ *are defined as in Eq.* (2)*, then with probability at least* $1/10$:

$$\forall t \in [T] : \quad \|w_t^S - w_t^{S'}\| \geq \Omega\left(\eta L \sqrt{t}\right).$$

One can then pick the dominant term between the bounds, and obtain that with probability at least $1/10$:

$$\|w_t^S - w_t^{S'}\| \geq \Omega\left(\frac{\eta L t}{\sqrt{n}} + \eta L \sqrt{t}\right). \tag{27}$$

Suppose some $u_t$, independent on the samples $S$ and $S'$. Then by the triangle inequality we have that,

$$
\begin{aligned}
\mathbb{P}\left(\|w_t^S - w_t^{S'}\| \geq a\right) &\leq \mathbb{P}\left(\|w_t^S - u_t\| + \|w_t^{S'} - u_t\| \geq a\right) \\
&\leq \mathbb{P}\left(\|w_t^S - u_t\| \geq a/2\right) + \mathbb{P}\left(\|w_t^{S'} - u_t\| \geq a/2\right) \\
&= 2\,\mathbb{P}\left(\|w_t^S - u_t\| \geq a/2\right). \qquad (S \text{ and } S' \text{ are i.i.d.})
\end{aligned}
$$

Dividing by 2 and using Eq. (27) we conclude the proof.

## C.1  Proof of Lemma C.1

Suppose $f : \mathcal{W} \times \mathcal{Z} \to \mathbb{R}$ takes following form:

$$f(w; z) = Lw \cdot z,$$

where $\mathcal{W} \subseteq \mathbb{R}$ and $z = \pm 1$ with probability $1/2$. Define a sample $S = \{z_1, \ldots, z_n\}$ and $S' = \{z'_1, \ldots, z'_n\}$, then by the update rule in Eq. (2) we obtain,

$$w_{t+1}^S = -\eta L t \cdot \frac{1}{n} \sum_{i=1}^{n} z_i, \quad w_{t+1}^{S'} = -\eta L t \cdot \frac{1}{n} \sum_{i=1}^{n} z'_i.$$

This implies that $|w_{t+1}^S - w_{t+1}^{S'}| = \eta L t \cdot |\frac{1}{n} \sum_{i=1}^{n} (z_i - z'_i)|$. Note that,

$$
z_i - z'_i = \begin{cases} 2 & w.p.\ 1/4, \\ 0 & w.p.\ 1/2, \\ -2 & w.p.\ 1/4. \end{cases}
$$

Using Berry-Esseen inequality one can show that with probability at least $1/10$:

$$\left|\frac{1}{n} \sum_{i=1}^{n} (z_i - z'_i)\right| \geq \frac{1}{\sqrt{n}}.$$

In turn we conclude that w.p. at least $1/10$,

$$|w_{t+1}^S - w_{t+1}^{S'}| \geq \frac{\eta L t}{\sqrt{n}}.$$

We remark that $f(w; z)$ can be embedded to any large dimension, thus implying our lower bound holds for any arbitrary dimension.

## C.2  Proof of Lemma C.2

This proof relies on the same construction of [5]. The difference is that we show a lower bound between iterates over two i.i.d. samples while their result holds for two samples that differ only on a single example. The main observation here is that with some constant probability, the problem is reduced to that of [5]. Consider the following $f : \mathcal{W} \times \mathcal{Z} \to \mathbb{R}$:

$$f(w; z) = -\frac{\gamma L}{2} zw + \frac{L}{2} \max_{i \in [d]} \{w_i - \varepsilon_i, 0\},$$

where $\mathcal{W} \subseteq \mathbb{R}^d$ and

$$
z = \begin{cases} 1 & w.p.\ 1/(n+1), \\ 0 & w.p.\ 1 - 1/(n+1). \end{cases}
$$

We also choose $\varepsilon_i$ such that $0 < \varepsilon_1 < \ldots < \varepsilon_d < \gamma\eta L/(2n)$ and a sufficiently small $\gamma = 1/(4\sqrt{dT})$, and $d > T$. Observe that for a given sample $S = \{z_1, \ldots, z_n\}$ the empirical risk is then,

$$F_S(w) = -\frac{\gamma L}{2n} \sum_{i=1}^{n} z_i w + \frac{L}{2} \max_{i \in [d]}\{w_i - \varepsilon_i, 0\}.$$

We now claim that with probability $(1 - \frac{1}{n+1})^n$ over $S'$, the empirical risk is given by,

$$F_{S'}(w) = \frac{L}{2} \max_{i \in [d]}\{w_i - \varepsilon_i, 0\}.$$

Conditioned on this event, we get that $\nabla F_{S'}(0) = 0$ and therefore $w_t^{S'} = 0$ for any $t \in [T]$. In addition, we know that the complementary event, namely, $z_i = 1$ for at least a single $i \in [n]$, is given with probability $1 - (1 - \frac{1}{n})^n$. Since,

$$\Big(1 - \frac{1}{n+1}\Big)^n \le (1 - \frac{1}{2n})^n \le e^{-1/2}, \quad \text{and} \quad \Big(1 - \frac{1}{n+1}\Big)^n \ge 1/2,$$

we have that with probability at least $0.5 \cdot (1 - e^{-1/2}) \ge 0.19$ both events occur. Note that $\nabla F_S(w) = -\frac{\gamma L}{2n} \sum_{i=1}^{n} z_i \mathbb{1} + \frac{L}{2} \nabla \max_{i \in [d]}\{w_i - \varepsilon_i, 0\}$ where $\mathbb{1}$ is the one vector. Then applying the update rule in Eq. (2) and the fact that $w_0^S = 0$ we get,

$$w_1^S = \frac{\gamma\eta L}{2n} \sum_{i=1}^{n} z_i \mathbb{1}.$$

Recall, that under the aforementioned event we have that $\frac{1}{n}\sum_{i=1}^{n} z_i \ge \frac{1}{n}$. This implies that $w_1^S(i) \ge \frac{\gamma\eta L}{2n} > \varepsilon_i$ for any $i \in [d]$. Therefore,

$$w_2^S = w_1^S - \eta \nabla F_S(w_1^S) = \frac{2\gamma\eta L}{2n} \sum_{i=1}^{n} z_i \mathbb{1} - \frac{\eta L}{2} e_1,$$

where $e_i$ is the standard basis vector of index $i$. Since $\gamma \le 1/(4T)$ we have that $w_2^S(1) \le \frac{\eta L}{4T} - \frac{\eta L}{2} < 0$. Developing this dynamic recursively we obtain,

$$w_{t+1}^S = \frac{\gamma\eta Lt}{2n} \sum_{i=1}^{n} z_i \mathbb{1} - \frac{\eta L}{2} \sum_{s \in [t]} e_s.$$

Using the reverse triangle inequality we have,

$$\begin{aligned}
\big\|w_t^S - w_t^{S'}\big\| = \big\|w_t^S\big\| && (w_t^{S'} = 0) \\
\ge \frac{\eta L}{2}\Big\|\sum_{s \in [t]} e_s\Big\| - \frac{\gamma\eta Lt}{2n}\Big|\sum_{i=1}^{n} z_i\Big|\|\mathbb{1}\| && \text{(reverse triangle inequality)} \\
\ge \frac{\eta L\sqrt{t}}{2} - \frac{\gamma\eta Lt}{2}\sqrt{d} && (\|\mathbb{1}\| = \sqrt{d} \text{ and } \Big|\sum_{i=1}^{n} z_i\Big| \le n) \\
\ge \frac{3}{8}\eta L\sqrt{t}. && (\gamma \le 1/(4\sqrt{dt})
\end{aligned}$$