# OpenReview forum: "Thinking Outside the Ball: Optimal Learning with Gradient Descent for Generalized Linear Stochastic Convex Optimization"
_NeurIPS.cc/2022/Conference — NeurIPS 2022 Accept_

### Official Review · Reviewer_VfrU · 2022-07-10

**Rating:** 6
**Confidence:** 4
**Soundness:** 4 excellent
**Presentation:** 3 good
**Contribution:** 3 good

**Summary:**

The paper studies the gradient descent for a stochastic convex optimization problem for a generalized linear model. With excess error at most $\epsilon$, sample complexity and number of iterations of $\tilde{O}(1/\epsilon^{2})$ is needed in contrast to the general stochastic convex optimization, where $\Omega(1/\epsilon^{4})$ is needed.

**Questions:**

(1) In the abstract, and introduction, you mentioned ''early stopped'' gradient descent, but when you introduce your gradient descent algorithm in (2) and afterwards, it seems to me (I hope I didn't miss anything) that you never talk about ''early stopped'' again. Where did you use ''early stopped'' in your analysis?

(2) In equation (3), should it be from $t=1$ to $T$?

(3) In equation (11), what is the space you are taking supremum over for $D$ and $f$?

(4) In line 478 in the supplementary material, it is better to use '','' instead of ''.'' to end the last equation and write ''where'' instead of ''Where'' here. Same can be said about line 495.

(5) Cauchy-Schwartz should be Cauchy-Schwarz.

(6) In the proof of Theorem 3.1, for the equation between line 494 and line 495, what do you mean by $P(w\notin\mathcal{W}_{u}^{K})$? The previous term involves taking sup over $w$, but here, it is for some arbitrary $w$? Also in the same line, what do you mean by $\sup_{S}$? What is the space of the supremum taken over? Can you explain in more detail how do you get the next inequality?

(7) Line 509. In the statement of Theorem, instead of ''Boucheron, Lugosi and Massart 7'' write ''Boucheron, Lugosi and Massart [7]''.

(8) In line 522, I am confused with the notation $u=\frac{1}{T}\sum_{t=1}^{T}u_{t}$. In the equation between line 262 and 263, you had $\bar{u}$. So both $u$ and $\bar{u}$ denote average?

(9) In line 528, can you provide a reference to McDiarmid or perhaps even state what it is?

(10) In line 531, line 538, what is ''union bound''?

(11) In line 533, ''set $w^{\ast}$'', do you mean ''set $w^{\star}$''?

(12) In line 545, maybe you can define Rademacher random variables for the convenience of the readers.

(13) In line 578, ''with the same probability'', I just want to know whether this is indeed with the same probability, or with at least the same probability?

(14) In line 600, what is ''reverse triangle inequality''? I know what triangle inequality is, but I am not familiar with reverse triangle inequality. Plus in the second line below, you wrote (triangle inequality) instead of (reverse triangle inequality).

**Limitations:**

It seems the author(s) did not include a discussions on the limitations and potential negative societal impact of their work.

**Strengths And Weaknesses:**

The logic and contributions of the paper are clear. To the best of my knowledge, the results are novel.

In terms of weakness, I think although the results are new, they are not that surprising because one knows that there is dimension-independent uniform convergence bounds for generalized linear models and it is reasonable to expect that sample complexity and number of iterations for a stochastic convex optimization problem for a generalized linear model will be better than the general stochastic convex optimization. In addition, I think the presentations can be improved here and there, and some points can be made more clear which I will detail in the questions below.

---

> ### Author Response · Authors · 2022-08-02
> **Thanks for the review**
>
> > “In terms of weakness, I think although the results are new, they are not that surprising because one knows that there is dimension-independent uniform convergence bounds for generalized linear models and it is reasonable to expect that sample complexity and number of iterations for a stochastic convex optimization problem for a generalized linear model will be better than the general stochastic convex optimization.”
>
> We were actually quite surprised to obtain this result due to recent works which showed a separation between GD and SGD. Note that unbounded-norm linear models do not have dimension independent uniform convergence. We also discuss in section 4 why the standard dimension-independent uniform convergence bounds fail here.  We then show how one can apply a distribution **dependent** uniform convergence argument (Thm 3.2) in this setup, and this was not trivial to us.
>
>
> > “In the abstract, and introduction, you mentioned ''early stopped'' gradient descent, but when you introduce your gradient descent algorithm in (2) and afterwards, it seems to me (I hope I didn't miss anything) that you never talk about ''early stopped'' again. Where did you use ''early stopped'' in your analysis?”
>
> The second part of Theorem 3.1 is achieved for $T=n$, when the optimization error reaches $O(1/\sqrt(n))$ and not $0$. Note that for $T\rightarrow\infty$ the bound would be vacuous. We will clarify this point in the updated version of the paper.
>
> > “In equation (3), should it be from $t=1$ to $T$?”
>
> Yes, thanks!
>
> > “In equation (11), what is the space you are taking supremum over for $D$ and $f$?”
>
> $D$ and $f$ are taken over all valid distributions such that $f$ is convex and lipschitz. $D$ is a probability measure (w.r.t Borel $\sigma$- algebra as is standard – we will note this).
>
> > “In the proof of Theorem 3.1, for the equation between line 494 and line 495, what do you mean by $P(w\notin\mathcal{W}_{u}^{K})$? The previous term involves taking sup over $w$, but here, it is for some arbitrary $w$? Also in the same line, what do you mean by $\sup{S}$? What is the space of the supremum taken over? Can you explain in more detail how do you get the next inequality?”
>
> It is a typo, it should read $\bar{w}^S$ instead of $w$. The supremum is taken over all possible samples (to be exact, and we will clarify, samples of $n$ norm bounded $x$’s and labels $y$).
>
> The inequality is given as follows: the first term of the LHS is bounded as stated (using Eq.(13) with $K$ being the RHS of Eq.(14))  and the second term is bounded by bounding the worst case norm of $w_S$ as described in the sentence below.
> Does that clarify this?
>
> > “In line 522, I am confused with the notation $u=\frac{1}{T}\sum_{t=1}^Tu_t$. In the equation between line 262 and 263, you had $\bar u$. So both $u$ and $\bar u$ denote average?”
>
> Yes, for clarity we will use a single notation.
>
> >  “In line 528, can you provide a reference to McDiarmid or perhaps even state what it is?”
>
> Sure. Frankly, the best reference here is wikipedia. But we will also add a reference in the paper to Lemma 26.4 in [33].
>
> [33] Understanding machine learning: From theory to algorithms by Shai Shalev-Shwartz et al.
>
> > “In line 531, line 538, what is ''union bound''?”
>
> The union-bound inequality states that for a sequence of events $A_1,..,A_n$: $P(\cup_{i=1}^n A_i)\leq \sum_{i=1}^nP(A_i)$.
>
> > “In line 578, ''with the same probability'', I just want to know whether this is indeed with the same probability, or with at least the same probability?”
>
> It is with at least the same probability. Thanks, we will fix it.
>
> > “In line 600, what is reverse triangle inequality?”
>
> The reverse triangle inequality states that $||x+y|| \geq \big|||x||-||y||\big|$

---

> > ### Comment · Reviewer_VfrU · 2022-08-09
> > **response to rebuttal**
> >
> > Thanks for the clarifications! Please make sure you incorporate all these changes in the revised version of the paper.

---

> ### Comment · Area_Chair_Vuse · 2022-08-08
> **Interacting with rebuttal**
>
> Dear reviewer,
>
> Can you read the author's rebuttal, check if it addresses your concerns, and react to it?
>
> It is important to acknowledge this work by the authors and to respect it.
>
> Best,
> AC

---

### Official Review · Reviewer_csuM · 2022-07-13

**Rating:** 6
**Confidence:** 3
**Soundness:** 4 excellent
**Presentation:** 3 good
**Contribution:** 2 fair

**Summary:**

The authors study the problem of excess risk minimization of convex Lipschitz Generalized Linear Models (GLMs) over euclidean norm balls.

They consider the classical setting of minimization of the empirical risk as a surrogate for the excess risk.
They consider the Gradient Descent (GD) algorithm to optimize the empirical risk, except they _do not_ rely on projections and simply run plain GD.

Using the uniform convergence of GLMs, the authors show that plain GD with early stopping outputs an $\epsilon$-accurate solution (compared to the best solution contained in a euclidean norm ball) in $O(1/\epsilon^2)$ iterations with $O(1/\epsilon^2)$ samples.

This contrasts with the performance of GD under the more general setting (i.e. non-GLM) where the complexity is $O(1/\epsilon^4)$.

Finally, the authors show through a lower bound and matching upper bound a non-distribution-dependent convergence is possible for GD without projections, but the resulting is then $O(1/\epsilon^4)$.



**Questions:**

Apart from the above questions, here are some more minor remarks:

* I did not fully understand the last part of the sentence line 126.
* In the second equation of Theorem 3.1, I believe $\norm{w}^2$ should be $B^2$
* Supplemental material, equation after line 503: the $c>0$ is refered to as $t$ and then $c$.
* Supplemental material, to apply Theorem line 509 to the current problem, I believe $c_i= 4L$, while the authors use $c_i = L$. This has no major impacts, but it changes the constants in the rest of the proof and in the theorem statement.
* Equation 10: $g(a) = ...$
* line 284: "the" -> "that"


**Strengths And Weaknesses:**

# Strength

The paper is well-written and easy to follow.
The problem at hand is well described, and even though I am not an expert in the field, the related work section seems complete.
The proofs are kept simple and arguments follow each other nicely.

I particularly appreciate part 5 which gives an overview of the main arguments of the proofs, and make diving in the proofs easier.

# Weaknesses
I am wondering about the relevance of the result.
Namely, as the authors mention, rates of $O(1/\epsilon^2)$ are already known for SGD, and SGD is the go-to method in practice as iterations are much cheaper than GD.
So is the main contribution that GD reaches that rate without projections ? or is it the distribution dependent uniform convergence argument ? or a mix of both ? I know the authors discuss this at the end of the introduction, but it would be great if they could clarify.

Also, the authors often refer to the notion of stability, especially at the end of section 5, but I believe that they never actually define it. Could the authors clarify this ?

---

> ### Author Response · Authors · 2022-08-02
> **Thanks for the review**
>
> > “I am wondering about the relevance of the result. Namely, as the authors mention, rates of $(1/\epsilon^2)$ are already known for SGD, and SGD is the go-to method in practice as iterations are much cheaper than GD. So is the main contribution that GD reaches that rate without projections? or is it the distribution dependent uniform convergence argument ? or a mix of both ?”
>
> On the technical side we believe that the distribution dependent uniform convergence argument we obtain here is indeed a contribution and may be of independent interest for future research.
>
> The main result of our paper is the analysis of GD without projections. Our starting point is that previous analysis (Amir et al.) showed that, in the convex setup, the extreme case of full-batch can indeed hurt the learner. The result here shows that this phenomenon (full-batch can hurt) does not happen anymore in the natural setup of GLMs, even without projections.
>
> We want to be very careful when interpreting the relevance of these results to practical setups: we would like to add the disclaimer that neither SGD or GD are used in practice but in fact mini-batches are the go-to method, and it is a very big question how batching the samples hurt or help the learner in more practical setups. Here we show that, at least for GLMs, taking full-batches doesn’t hurt (even though, more generally, it can hurt).
>
> > “Also, the authors often refer to the notion of stability, especially at the end of section 5, but I believe that they never actually define it. Could the authors clarify this?”
>
> In section 5 we refer to the notion of uniform argument stability defined in Bassily et al. We will clarify this and make it explicit:
> It is the difference between the algorithm trajectories of $w^S_t$ and $w^{S’}_t$ over samples $S$ and $S’$ that differ on a single example, e.g. $S=(z_1,\ldots,z_i,\ldots,z_n)$ and $S’=(z_1,\ldots,z’_i,\ldots,z_n)$.
>
> > “I did not fully understand the last part of the sentence line 126.”
>
> Thanks for pointing that out, we will clarify this sentence. We suggest the following revision:
>
> though it appears to be suboptimal, distribution independent uniform convergence can explain generalization even though the norm could become very large (by making explanations based on uniform convergence inside a fixed distribution independent class)
>
> > “In the second equation of Theorem 3.1, I believe $||w||^2$ should be $B^2$”
>
> Since the first equation holds for any $B$, you can obtain the result in the second equation of Theorem 3.1.
>
>
> > minor remarks and typos
>
> Thanks, we will fix those.

---

> > ### Comment · Reviewer_csuM · 2022-08-07
> > **Answer to the rebuttal**
> >
> > Dear authors,
> >
> > I acknowledge your rebuttal and thank you for your helpful clarifications.
> >
> > As I mentioned previously, I believe this is a technically solid, well-written paper. However, I still have some concerns about what seems like a moderate contribution to the literature. I will therefore keep my score of 6.

---

### Official Review · Reviewer_JR1r · 2022-07-20

**Rating:** 6
**Confidence:** 3
**Soundness:** 2 fair
**Presentation:** 3 good
**Contribution:** 3 good

**Summary:**

This paper shows that gradient descent with early stopping in the context of Generalized Linear models has an optimal sample and iteration complexity.

**Questions:**

1. Some emphasis is put on the fact that this work is dealing with Gradient descent without projection, but the generalization guarantees are obtained with respect to the best classifier in a constrained ball $\mathcal{W}^B$. (see, for instance, Theorem 3.1). (in comparison, [2] uses projections).

2. In the same vein, I am a bit confused with some statements that compare the output of GD (unconstrained) and the solution of the constrained problem. For instance in (4) if we assume that the optimum is at $w=0$ and we take $B=\eta \approx 0$, we get that $F_S(\bar w^S) \leq \min_w F_S(w)$ which is not possible if we initialize at $w_0 \neq 0$ because we would have $\bar w^S \approx w_0$ and thus  $F_S(\bar w^S) \gg \min_w F_S(w)$. Can you comment on that and/or provide a precise citation/proof for equation (4)? I read [17] and the closest statement to (4) in [17] is Theorem 3.1 about online gradient descent where there is a projection step for the iterates. If we considered GD with projection, it would not be a problem since one projection step would bring us to the optimum. It seems to me that usually, in the proof of (4) the quantity $\|x^* - x^t\|$ appears and is upper bounded by $2B$ because $x^*$ and $x^t$ **belong to the constraint set.**
3. By reading the statement of Theorem 3.1, I was not 100% sure how you would go from the constrained to the penalized formulation. I got it by reading the proof of Thm 3.4. You use the fact that the constrained statement is valid for any $B$. Thus it is critical to make sure that (4) is valid and precise that the first statement of Theorem 3.1 is valid for any $B$.

4. In any case,  you could also assume that the optimum is achieved and use the fact that $$F_S(\bar w^S) \leq \min_{w \in \mathbf{R}^d} F_s(w) + \eta L^2 + \frac{\|w^*-w_0\|^2}{\eta T}$$
 ? Another alternative for (4) to be true would be to use $\mathcal{W}^B$ with $B$, and $u$ picked such that we know that $\bar w^S$ belongs to $\mathcal{W}^B$, but you would not get the penalized version of the result.


Overall, I may change my grade depending on the answers to these questions. (mainly question 2.) Precisely, I am currently leaving them the benefit of the doubt, but if the statement in Eq.4 is not clarified, I may decrease my score to 4. On the other hand, if all my questions are correctly addressed, I may increase my score to 7.


**Limitations:**

The iteration complexity is maybe optimal, but the oracle complexity is n * iteration complexity which is $ O(1/\epsilon^4)$. So still worse than SGD. It is briefly addressed by the authors.

**Strengths And Weaknesses:**

I am not an expert in the specific field of this paper, so it is hard for me to assess if this paper is not missing some important related work. However, to the extent of my knowledge, it seems that this paper is not missing any critical related work.

1. (Pro and con) On the one hand, the result seems new and interesting as it contrasts previous results on generalization of GD and SGD with early stopping. On the other hand, it seems that SGD is superior to GD even in that setting (since an iteration is significantly cheaper for SGD.) Thus we may consider that this result is not groundbreaking.  However, I still think this paper is in the interest of the community (assuming that its results are correct)

2. (Pro) The core of the technical contribution (Thm 3.1) is very well explained.

3. (Con) I found the exposition of some of the results not 100% clear. I have some serious doubts about some results mentioned in this paper (mainly equation (4) and its consequences). I am hopeful that the authors can clarify this during the rebuttal. (see my questions)

---

> ### Author Response · Authors · 2022-08-02
> **Thanks for supporting acceptance**
>
> > “Some emphasis is put on the fact that this work is dealing with Gradient descent without projection, but the generalization guarantees are obtained with respect to the best classifier in a constrained ball $\mathcal{W}^B$”
>
> Limiting the generalization guarantee against a best classifier in a constrained ball is standard in this setup. In fact, even for plain optimization problems, without the question of generalization.
> Observe that, given a finite number of gradient steps, one cannot hope to obtain a meaningful upper bound on the difference between the optimization loss and the loss of the best hypothesis in the entire space $\mathbb{R}^d$. That is why it is common to compare ourselves to the best hypothesis in some bounded ball.
>
> > “I am a bit confused with some statements that compare the output of GD (unconstrained) and the solution of the constrained problem … Can you comment on that and/or provide a precise citation/proof for equation (4)?”
>
> Equation (4) is valid as stated. First, notice that we consider GD that is initialized at $w_0=0$. We state this right after Eq. (2), and it is a very standard assumption in the context of convex optimization (this is basically without any loss of generality and the general result should be interpreted accordingly – namely, the ball is centered around the initialization). We will make sure to emphasize this assumption.
>
> A more precise citation: The proof of Eq. (4) is indeed extremely similar to the proof of Theorem 3.1 in [17]. However, you are right that [17] assumes projections and we should have provided a better reference. We thank the reviewer for noticing this.
> A better reference is [33] where Eq. (4) follows immediately from Lemma (14.1) (Eq. (14.5)+Eq. (14.3) to be exact) and Corollary 14.2 exemplifies it for a specific learning rate.
>
> We will provide this citation instead and also emphasize the initialization at $0$. Besides changing the reference, we will add a complete proof of Eq. (4) to the appendix along the above lines (for completeness).
> We hope this concern is addressed properly, if you still find it unclear we would gladly clarify it further in the discussion period.
>
> [33] Understanding Machine Learning: From Theory to Algorithms by Shai Shalev-Shwartz et al.
>
> > “By reading the statement of Theorem 3.1, I was not 100% sure how you would go from the constrained to the penalized formulation. I got it by reading the proof of Thm 3.4. You use the fact that the constrained statement is valid for any $B$. Thus it is critical to make sure that (4) is valid and precise that the first statement of Theorem 3.1 is valid for any $B$
>
> You are right, a quantifier is missing over $B$ in the constrained statement. We should add it to the constrained statement “for every B”. Thanks!!!
> And you are also right that it is crucial that eq. 4 holds for every $B$.
>
> > “On the other hand, it seems that SGD is superior to GD even in that setting (since an iteration is significantly cheaper for SGD.) Thus we may consider that this result is not groundbreaking. However, I still think this paper is in the interest of the community (assuming that its results are correct)”
>
> We agree that in terms of gradient computations SGD is superior to GD, and we don’t claim otherwise in the paper. Nonetheless, GD is still one of the most well studied algorithms in the machine learning literature and it serves as an ideal candidate for furthering our understanding of generalization in different regimes.

---

> > ### Comment · Reviewer_JR1r · 2022-08-08
> > **Thank you for your answer**
> >
> > Thanks for your answer.
> >
> > You have clarified all my concerns except this one
> >
> > > Limiting the generalization guarantee against a best classifier in a constrained ball is standard in this setup. In fact, even for plain optimization problems, without the question of generalization.
> >
> > Since it is standard, can you provide some standard references?
> > Preferably examples for which the algorithm's output (in your case $\bar w_T$) is not bounded by the same bound as $\mathcal W^B$.
> >
> > Also, can we hope for better than $\| \bar w_T\|  \leq O(LT)$?

---

> > > ### Author Response · Authors · 2022-08-09
> > > **Thank you for responding**
> > >
> > > > Since it is standard, can you provide some standard references?
> > >
> > > In [33], Lemma 14.1 you can find such an example where the algorithm output is not necessarily bounded. Specifically, they consider gradient steps without projection, similarly to our work. Another work that demonstrates this is [37], Corollary 4 where the reference point $w_o$ is norm bounded but the algorithm output, namely $\tilde w$, is not.
> > >
> > > [33] Understanding Machine Learning: From Theory to Algorithms by Shai Shalev-Shwartz and Shai Ben-David.
> > >
> > > [37] Fast Rates for Regularized Objectives by Karthik Sridharan, Shai Shalev-Shwartz and Nathan Srebro.
> > >
> > > The more general claim we tried to convey here, is that it is standard in generalization problems, to compare oneself to a reference point in a bounded domain. Another example that exhibits this can be found in [41], Theorem 3.4.
> > >
> > > [41] Introduction to Online Convex Optimization by Elad Hazan.
> > >
> > > > Also, can we hope for better than $\|\bar {w}_T\|\leq O(LT)$?
> > >
> > > Considering a worst case analysis, one cannot hope to obtain a better bound on the norm. For example, the output of GD on a linear lipschitz function $f(x)=L\cdot x$ will obtain the norm $\Theta(\eta L T)$.
> > >
> > > We hope to have addressed your questions properly and we appreciate your feedback.

---

> > > > ### Comment · Reviewer_JR1r · 2022-08-09
> > > > **Thank you**
> > > >
> > > > thank you for your answers.
> > > >
> > > > I maintain my score

---

> ### Comment · Area_Chair_Vuse · 2022-08-08
> **Interacting with rebuttal**
>
> Dear reviewer,
>
> Can you read the author's rebuttal, check if it addresses your concerns, and react to it?
>
> It is important to acknowledge this work by the authors and to respect it.
>
> Best,
> AC

---

### Comment · Area_Chair_Vuse · 2022-08-03
**Discussion period**

Thanks to all reviewers and authors for their work on this submission.

As the discussion period starts, I want to make sure that reviewers have read the author's response, and if needed react to it.

This can be done either by communicating with authors or in private conversation within the reviewing team.

---

### Meta-Review · Area_Chair_Vuse · 2022-08-26

**Recommendation:** Accept
**Confidence:** Certain

**Metareview:**

This work is concerned with an analysis of early-stopped gradient descent in a stochastic setting, and shows improved complexity, by relying on an assumption of a distribution-dependent ball for the parameter.

The reviewers all think that this is an interesting submission that should be accepted, and I agree with them.

**Award:**

No

---

### Decision · Program_Chairs · 2022-09-14

Accept